# The first complete hand-rearing of two neonatal finless porpoises

**Masahiko Kasamatsu[1], Kazuhiro Hasegawa[1], Ikuo Wakabayashi[1], Masami Furuta[1], Hiroki Inoue[2], Hidetomo Iwano[2]\***

1 Marine Biological Laboratory, Toba Aquarium, Toba, Japan, 2 Laboratory of Veterinary Biochemistry, School of Veterinary Medicine, Rakuno Gakuen University, Ebetsu, Hokkaido, Japan

\* h-iwano@rakuno.ac.jp

**Data Availability Statement:** All relevant data are within the manuscript and its Supporting information files.

**Funding:** The authors declare that no specific funding was received for this work.

## Abstract

Hand-rearing of marine mammals is an essential technique for the husbandry of orphans in captivity or the wild, especially endangered cetacean species. The purpose of the present study was to establish a method for successful hand-rearing and evaluate the nutritional state of neonatal finless porpoises. Two neonate finless porpoises maternally neglected at 5 days of age (Day 5) (neonate A, animal A) and Day 4 (neonate B, animal B) were hand reared. The amount of each tube feeding and daily number of nursings for animals A and B during the lactation period were gradually increased to 1,355 and 1,120 ml and 16 and 14 times, respectively. The mean daily caloric intake during the lactation period and average increase in body weight of animals A and B were 2,048 ± 207 and 2,206 ± 169 kcal and 65.4 and 66.9 g/day, respectively. Hypoproteinemia and hypertriglyceridemia were observed in the two neonates during the early stage of hand-rearing. The plasma concentrations of 24 free amino acids in the neonatal porpoises were significantly higher compared with adult porpoises. Plasma valine, leucine, and isoleucine levels in the neonates were dramatically higher than those in adults. Hyperlipoproteinemia, characterized by a higher percentage of very-low-density lipoprotein and the appearance of midband, was also observed in the two neonates, along with hypertriglyceridemia. A hand-rearing method for finless porpoises was successfully established in this research. Nutritional evaluation of serum protein, free amino acids, and lipid components is needed to improve the survivability of hand-reared neonatal porpoises. The hand-rearing method established in the present study is an essential technique for the husbandry of finless porpoises and can be applied to the conservation of other members of the porpoise family, including vaquita and Yangtze finless porpoises, which are the most endangered dolphins in the world.

## Introduction

Hand-rearing of marine mammals is an essential technique for the husbandry of orphans in captivity or the wild and for their conservation, especially with regard to endangered cetacean species. However, hand-rearing of cetacean calves is difficult. The first recorded successful hand-rearing of an infant harbor porpoise (*Phocoena phocoena*) occurred at the Point Defiance

**Competing interests:** The authors have declared that no competing interests exist.

Zoo and Aquarium, Washington, USA, and that of a bottlenose dolphin (*Tursiops truncatus*) took place at the Gulfarium, Florida, USA, in 1989 [1]. The difficulty of hand-rearing dolphins can be attributed to the large number of breast feedings necessary and the lack of information regarding artificial formulas.

Artificial formula that matches the physiological characteristics of dolphins, in conjunction with optimizing the amount of formula fed at each lactation and the daily number of nursings, are essential for the successful hand-rearing of dolphins. In order to develop an optimal artificial formula feed for dolphins, information regarding various marine mammal formula feeds that have been reported was organized. Two types of artificial formula have been reported and used in the hand-rearing of bottlenose dolphin, harbor porpoise, killer whale (*Orcinus orca*), pygmy sperm whale (*Kogia breviceps*), Risso's dolphin (*Grampus griseus*), and spotted dolphin (*Stenella attenuata*) [1–3]. Safflower oil in the April formula and salmon oil in the Sea World formula have been used for the hand-rearing of dolphins, but whipping cream has only been used in the Sea World formula [1–3]. In these cases, dolphin artificial formulas have been supplemented with herring filets with viscera, oil (salmon oil or safflower oil), and heavy whipping cream to increase the caloric density [1–4]. Although the fat content of bottlenose dolphin milk is 26.5%, Townsend reported that the fat content of artificial milk does not need to match the high fat level of mother's milk [1,2]. However, there is no indicator regarding how much lipid should be included in artificial milk. Townsend also described methods for the delivery of artificial formula to dolphin calves [1,2]. The optimal method for bottle feeding was for calves to suckle the artificial formula spontaneously from an artificial nipple. A tube feeding technique using a soft rubber stomach tube with a rounded tip was also useful for hand-rearing harbor porpoises.

Feeding frequency depends upon the caloric density of the formula, daily calorie requirements, and maximum volume of formula per feeding [2]. Combined knowledge regarding the approximate caloric density of the formula and the calorie requirements of calves allow calculation of the total daily volume of formula. Daily calorie requirements have been reported for several species of dolphins, harbor porpoises, and whales [2,3].

Hematology and serum chemistry techniques are essential tools for assessing the health status of dolphins [5]. Frequent blood examinations should be performed in neonates because hematological and biochemical values in dolphins are often the earliest indicators of subclinical disease [1]. Twice-weekly blood sampling and daily weighing of calves for 2 weeks are recommended until the calf is medically stable [2]. It is important to understand the physiological aspects of carbohydrate, protein, and lipid metabolism in dolphins for successful hand-rearing.

In the case of hand-rearing of dolphins, monitoring of serum protein, lipid components, and lipoproteins in calves is necessary for nutritional evaluation. Several recent reports have described methods for hand-rearing of dolphins, formula selection and preparation, artificial formula delivery, calorie requirements, and growth of neonates [1–3]. However, other than parameters pertaining to the formula, calorie requirements, and neonatal growth, little is known about analyses that could promote the success of hand-rearing of neonates from a few days after birth to the day of weaning. Few studies have focused on nutritional monitoring to determine health status or the normal growth of neonatal dolphins. Miyaji reported that although plasma amino acid concentrations vary under different physiological conditions, monitoring their concentrations could be a useful tool for assessing the health status of cetaceans [6].

The finless porpoise (*Neophocaena asiaeorientalis*) is a small member of the toothed whale family and is listed as an endangered species along with all other cetaceans [7,8]. The finless porpoise is distributed throughout tropical and subtropical coastal waters from the Persian

Gulf to Japan [7], but it is threatened by habitat degradation, water pollution, and bycatch. Many porpoise orphans rescued from the wild have not survived over the last few decades because no suitable hand-rearing techniques have been established. The establishment of hand-rearing techniques for endangered dolphins, particularly the analysis of artificial formulas and nutritional evaluation of neonatal dolphins, is of paramount importance for the conservation of wild dolphins, including their *ex situ* conservation in aquariums. The purpose of the present study was to establish a method for successful hand-rearing of finless porpoises and evaluate the nutritional status of neonate porpoises.

## Materials and methods

### Ethics statement

The objective of the hand-rearing techniques assessed in this study was to enable the rescue of orphaned endangered porpoises, and ethical approval was granted by the director of Toba Aquarium as an emergency measure (permit M-07-2013). The husbandry and animal care protocol used in this research complied with the current laws of Japan, the country in which the study was performed. The study was conducted in accordance with the guidelines of the Japanese Society of Veterinary Science and Japanese Society of Zoo and Wildlife Medicine for the ethical use of animals in research. Ethical considerations and data collection methods were approved by the responsible curator and director of the Toba Aquarium. Procedures for animal husbandry, veterinary care, and welfare of the study species in the Toba Aquarium strictly adhered to JAZA (Japanese Association of Zoos and Aquariums) standards for the accommodation and care of animals in zoos and aquariums.

### Animals

Two neonate finless porpoises were born at Toba Aquarium on May 2, 2013, and July 3, 2014. In the case of animal A, the first nursing was started 5 hours after birth. However, maternal neglect occurred at 5 days of age (Day 5). Tube feeding of animal A was started on Day 6. The mother and animal A were kept in the same exhibition pool until Day 6, and then animal A was removed from the mother and hand-reared from Day 7 onward in the medical pool. The body weight and body length of animal A on Day 5 were 9.83 kg and 76.5 cm, respectively. Animal A was force-fed whole fish beginning on Day 88 and then completely weaned by Day 136. The lactation period of animal A was between Days 6 and 87. The pre-weaning period was from the start of force-feeding of whole fish until the calf was weaned.

The first nursing of animal B was started 8 hours after birth. When physical and blood examination of animal B was performed on Day 4, swelling of the palatine mucosa of the maxilla and a comminuted fracture of the maxilla edge were observed. Animal B was removed from the mother and hand-reared from Day 4 in the medical pool. The body weight and body length of animal B on Day 4 were 9.76 kg and 88.5 cm, respectively. Animal B was force-fed beginning on Day 83 and completely weaned by Day 138.

### Initial feeding and formula

We tube-fed the two neonates an artificial milk formula through a stomach tube inserted directly into the stomach. A 60-cm stomach tube and 50-ml syringe were used for tube feeding in this study.

The formula was composed of powdered milk substitute (Puppy milk replacers, Pet-AG, Inc., Hampshire, IL 60140), whipping cream, salmon oil, slurry of fish meat (fish paste), multi–amino acids (Amiparen, Otsuka Pharmaceutical Co., Ltd., Chiyoda-ku, Tokyo, Japan),

**Table 1. Ingredients in formula for hand-rearing of finless porpoises.**

| Milk components | | | |
| --- | --- | --- | --- |
| **Ingredients** | **Initial** | **Lactation** | **Pre-weaning** |
| | **Day 1—Day 10** | **Day 11—Day 82** | **Day 83—Day 137** |
| Milk substitute | 30—60 ml [a] | 70—85 ml [a] | 50—150 ml [b] |
| Heavy whipping cream | 4 ml | 4 ml | 0—4 ml |
| Salmon oil | 2—4 ml | 2 ml | 0—3 ml |
| Multi-amino acid [c] | 0 ml | 10 ml | 0 ml |
| **Fish meat components** | | | |
| Slurry of fish meat | 0 ml | 5—45 ml | 0—150 ml |

Initial, the first ten days of hand-rearing; Lactation, neonates fed only artificial formula; Pre-weaning, neonates fed artificial formula and whole fishes.

[a] 100 g powdered milk (Puppy milk replacers, Pet-AG, Inc., 255 Keyes Avenue, Hampshire, Illinois 60140) added into 300 ml hot water.

[b] added into 400 ml hot water.

[c] Amiparen (Otsuka Pharmaceutical Co., Ltd., Chiyoda-ku, Tokyo, Japan) added during four and seven weeks of age.

and water (Table 1). Powdered milk substitute and water were mixed in a commercial blender, and then whipping cream, salmon oil, multi–amino acids, multi-vitamins, and lactoferrin (Morinaga Milk Industry Co., Ltd., Minato-ku, Tokyo, Japan) were added to the blended milk substitute right before tube feeding. The slurry of fish meat was made from strained, blended fish meat. Fish meat in which the bones and water had been removed were mixed to a uniform homogenate in a commercial blender and then strained trough a fine-mesh net to remove pieces of bones and scales. The slurry of fish meat was tube-fed to the neonates without mixing into the prepared milk substitute. Each neonate was supplemented with one-half of a multi-vitamin tablet (5M26 Vita-Zu tablet, Mazuri, IN, USA), a lactobacillus-based probiotic (half tablet, BID, Biofermin-R, Takeda Pharmaceutical Co., Ltd.), and 250 mg lactoferrin (SID) perday. Each multi-vitamin tablet contained retinol (10,000 IU), vitamin D (2000 IU), α-tocopherol (40 mg), acetomenaphthone (5 mg), thiamine (120 mg), riboflavin (8 mg), pyridoxin (5 mg), calcium pantothenate (10 mg), ascorbic acid (150 mg), nicotinamide (20 mg), cyanocobalamin (0.015 mg), choline (20 mg), inositol (5 mg), folic acid (1 mg), biotin (0.05 mg), and iron (12 mg).

## Weaning procedures

The weaning process for animals A and B was started on Day 88 and Day 83, respectively. During the pre-weaning period, the daily number of nursings and quantity of total tube feeding were gradually reduced around Day 90. Animals A and B were force-fed silver-stripe round herring (*Spratelloides gracilis*) and Japanese jack mackerel (*Trachurus japonicus*) during the pre-weaning period. Animals A and B were completely weaned by Day 136 and Day 138, respectively, and the total pre-weaning period lasted 88 days and 55 days, respectively. The amount of daily diet was increased from 6 g to 2,024 g in animal A and from 6 g to 2,275 g in animal B during the pre-weaning period.

## Blood samples

Blood samples were drawn into evacuated tubes from the tail fluke of the porpoises. Blood was collected into a serum separator tube (TERUMO, Tokyo, Japan) and an ethylenediamine

tetraacetic acid tube (TERUMO, Tokyo, Japan) for serum chemistry and complete blood count analysis, respectively. Hematology and serum chemistry values were determined within 12 h. The blood samples from both animals were centrifuged at 1600 × $g$ after coagulation, and the collected serum was immediately used for biochemical analysis.

### Analysis

White blood cell count, red blood cell count, hemoglobin level, and hematocrit level were determined using an automated analyzer (CELL-DYN 3200, Abbott, IL, USA). The concentrations of serum chemistry analytes in neonatal samples were determined using a TBA™-c16000 automated analyzer (Toshiba Medical Systems, Tochigi, Japan) and included analyses for total protein, total cholesterol, triglycerides (TGs), phospholipids, free fatty acid (FFAs), glucose, and blood urea nitrogen [5]. Serum lipoproteins were analyzed by polyacrylamide gel electrophoresis using a lipoprotein test kit, as described previously [9,10].

The concentrations of amino acids in plasma were measured using high-performance liquid chromatography–electrospray ionization–mass spectrometry (LC-MS/MS), followed by precolumn derivatization according to the method of the EZ:faast amino acid testing kit (Phenomenex, CA, USA) [11]. LC-MS/MS was carried out using a Prominence high-performance liquid chromatography system with an MS8030 tandem mass spectrometer (Shimadzu, Kyoto, Japan). In the present study, we focused on 40 amino acids: arginine, glutamine, citrulline, anserine, serine, asparagine, prolyl-hydroxyproline, hydroxyproline, 3-methylhistidine (3-MH), 1-methylhistidine, glycine, glycyl-proline, threonine, beta-alanine, alanine, sarcosine, hydroxylysine, gamma-aminobutyric acid, beta-aminoisobutyric acid, alpha-aminobutyric acid, ornithine, carnosine, methionine, proline, lysine, aspartic acid, histidine, thioproline, valine, glutamic acid, tryptophan, alpha-aminoadipic acid, phenylalanine, leucine, isoleucine, aminopimelic acid, cystathionine, cysteine, tyrosine, and kynurenine.

### Statistical analysis

The amount of tube feeding, daily calorie intake, hematological and serum biochemistry values, concentrations of serum lipid components, and plasma free amino acid concentrations in the two neonates are presented as mean ± SD. Differences in daily calorie intake and plasma free amino acid levels were analyzed using the Student's $t$-test. The null hypothesis was rejected at $P<0.05$.

### Results

To establish a method for successful hand-rearing, the two neglected neonate finless porpoises were hand-reared at 5 days (animal A) and 4 days (animal B) of age. The amount of each tube feeding and the daily number of nursings for animal A were gradually increased to 1,355 ml and 16 times, respectively, by Day 18 (Fig 1). During the lactation period from Days 19 to 87, the daily number of nursings for animal A was 13 or 14, and animal A gradually grew, consuming an average of 1,307 ± 52 ml of formula per day. Because frequent regurgitation was observed in animal A from Day 98, the amount of each tube feeding was increased, and the daily number of nursings was markedly decreased. On Day 100, endoscopy revealed mild esophagitis in the cardia of animal A. The daily number of nursings for animal A was gradually decreased during the pre-weaning period, and animal A was weaned on Day 135.

The amount of each tube feeding and daily number of nursings for animal B were gradually increased to 1,120 ml and 14 times, respectively, by Day 24 (Fig 1). During the lactation period, the daily number of nursings for animal B was between 10 and 14, and animal B gradually grew, consuming an average of 1,193 ± 64 ml of formula per day from Days 25 to 82. The

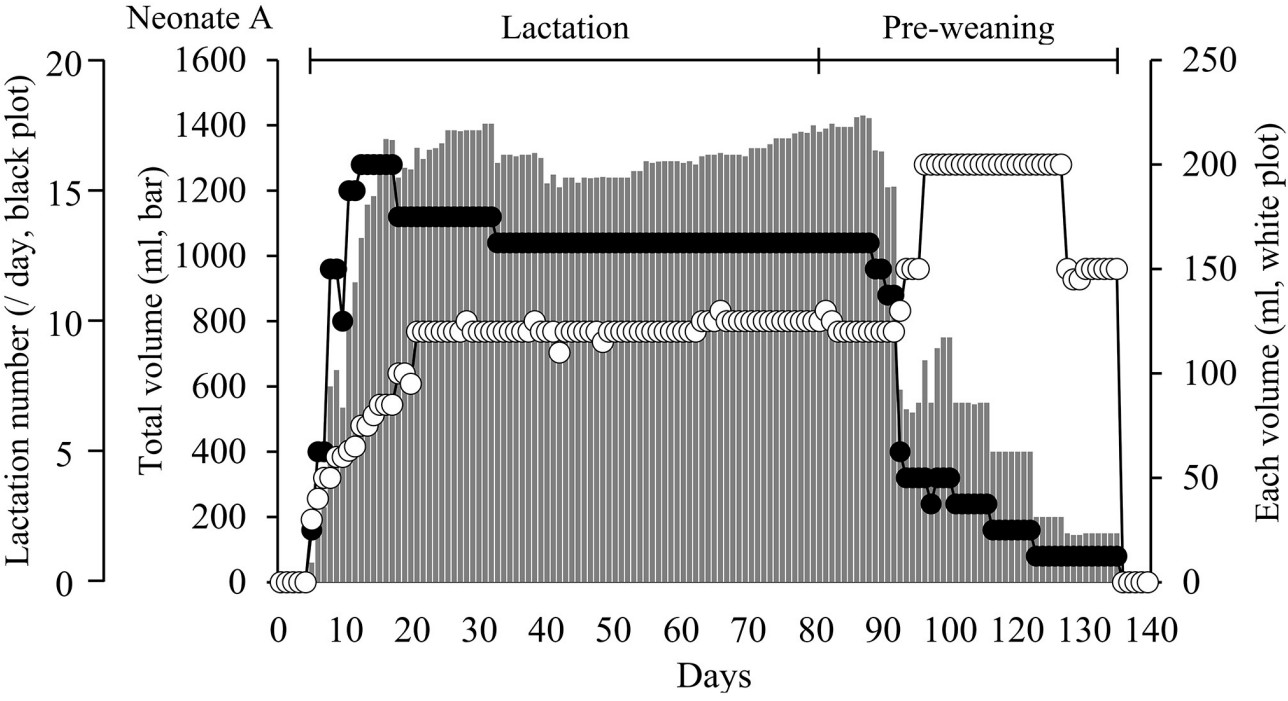

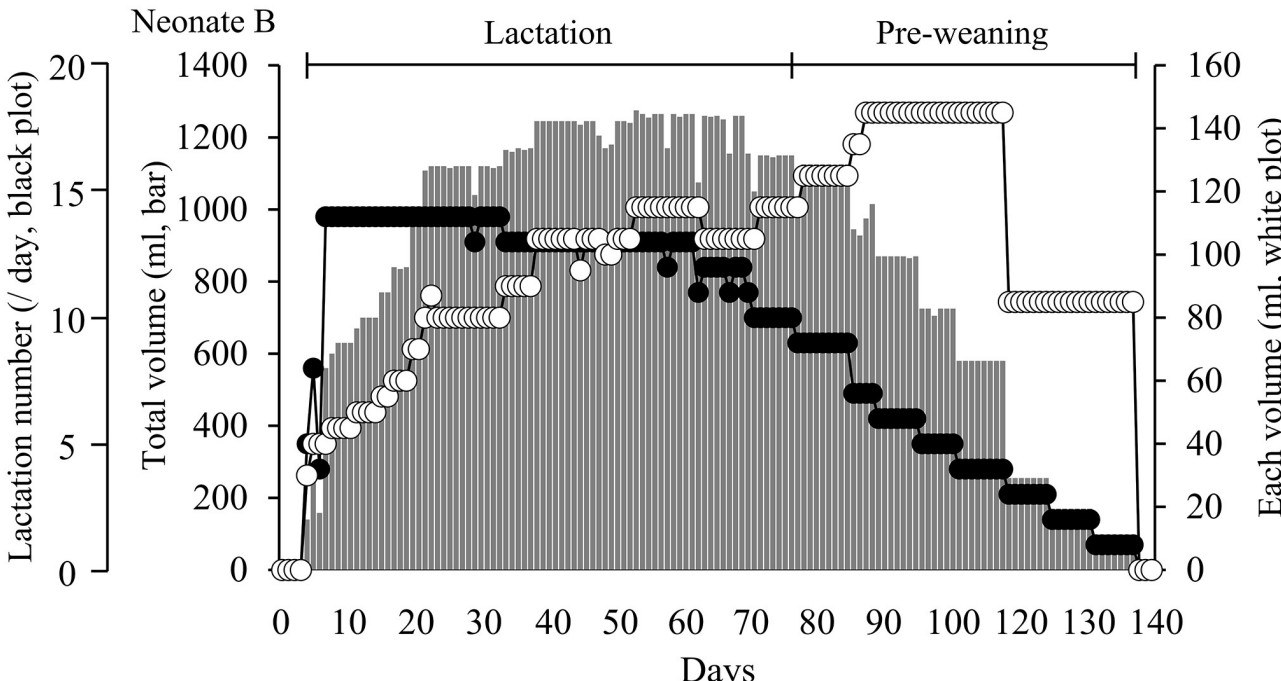

**Fig 1. Amount of each lactation, daily number of nursings, and total daily volume of formula for two hand-reared neonatal porpoises.** Upper panel shows data for animal A, and lower panel shows data for animal B. White data points indicate the amount of each lactation (vertical axis on the right side of the graph). Black data points indicate daily number of nursings (vertical axis on the left side of the graph). Bars indicate the total daily volume of formula (vertical axis on the left side of the graph).

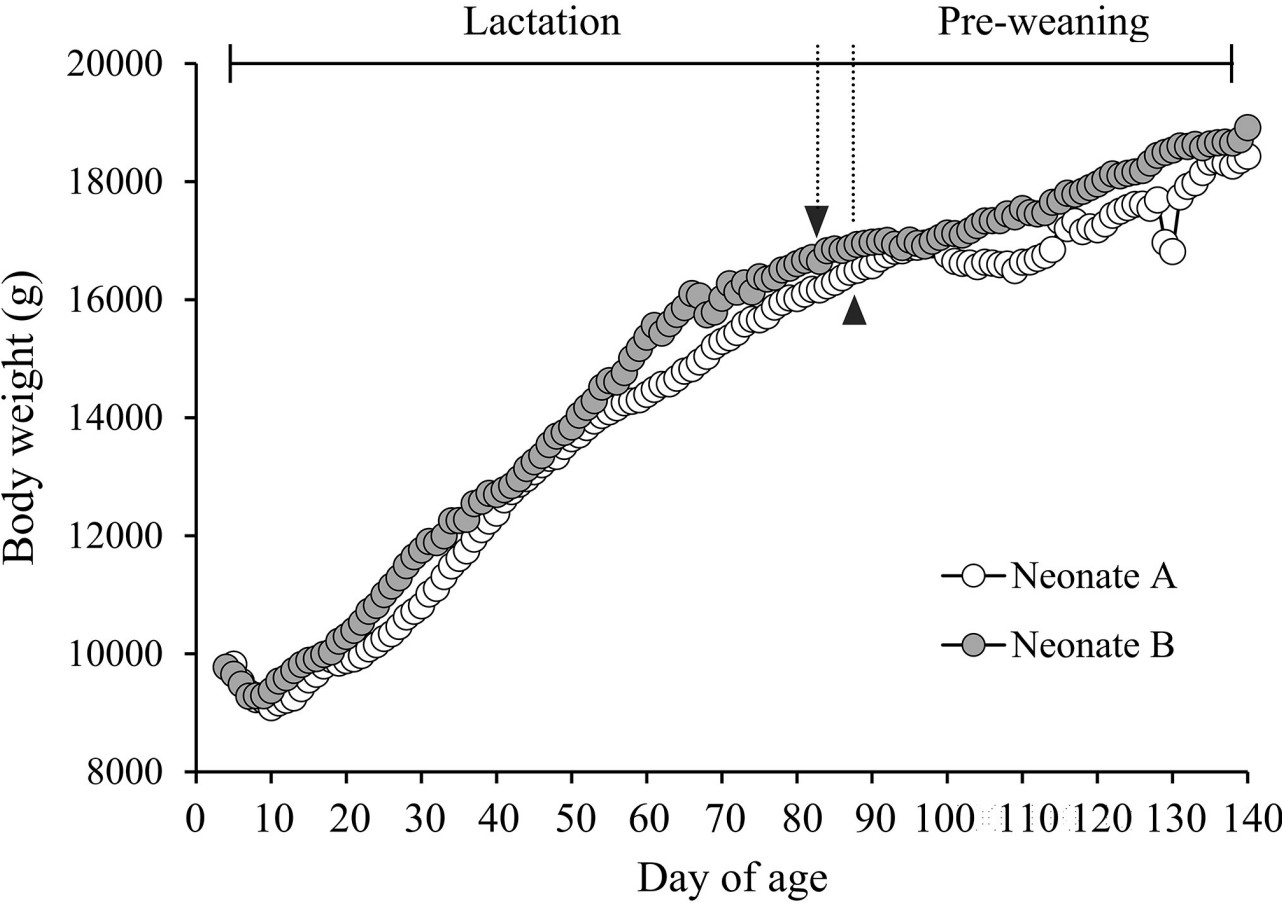

**Fig 2. Growth of two neonate porpoises during the hand-rearing period.** Lactation, neonates fed only artificial formula; Pre-weaning, neonates fed artificial formula and whole fish. White data points show data for animal A, and gray data points show data for animal B. Arrowheads indicate the day the weaning process was started.

weaning process for animal B was started on Day 83, and the daily number of nursings was gradually decreased during the pre-weaning period. Animal B was weaned on Day 137.

The weights of animal A and animal B are plotted in Fig 2. Animals A and B grew from a Day 5 weight of 9.83 kg to 18.33 kg at weaning (Day 136) and from a Day 4 weight of 9.78 kg to 18.67 kg at weaning (Day 138), respectively, during the hand-rearing period. The average increase in body weight for animals A and B was 65.4 and 66.9 g/day, respectively. The average increase in body weight for the two neonate porpoises during the pre-weaning period was 38.2 g/day, which was lower than that during the lactation period (85.0 g/day) in this study.

The daily calorie intake for animal A was gradually increased to 2,594 kcal over the first 14 days. The daily calorie intake for animal A was then decreased from Day 15 to Day 22 due to hyperlipidemia (Fig 3). The mean daily calorie intake for animal A was 2,048 ± 207 kcal during the lactation period from Days 23 to 87. On Day 88, animal A exhibited frequent regurgitation, and therefore, the number of nursings and daily calorie intake were decreased. The daily total calorie intake for animal A during the pre-weaning period was significantly (*P*<0.05) higher than that during the lactation period.

The daily calorie intake for animal B was gradually increased to 2,004 kcal over the first 21 days. The average daily calorie intake for animal B was 2,206 ± 169 kcal from Days 21 to 82

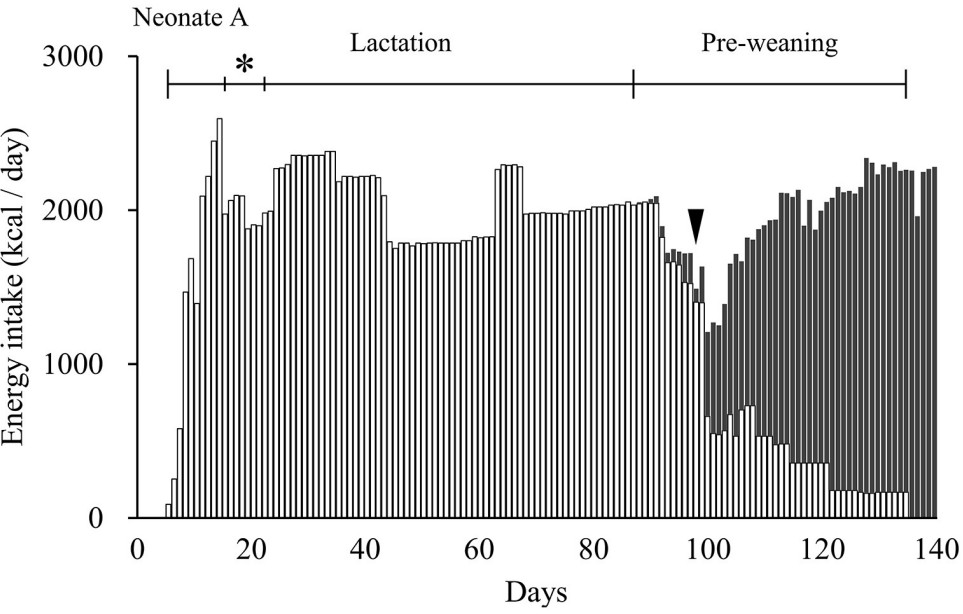

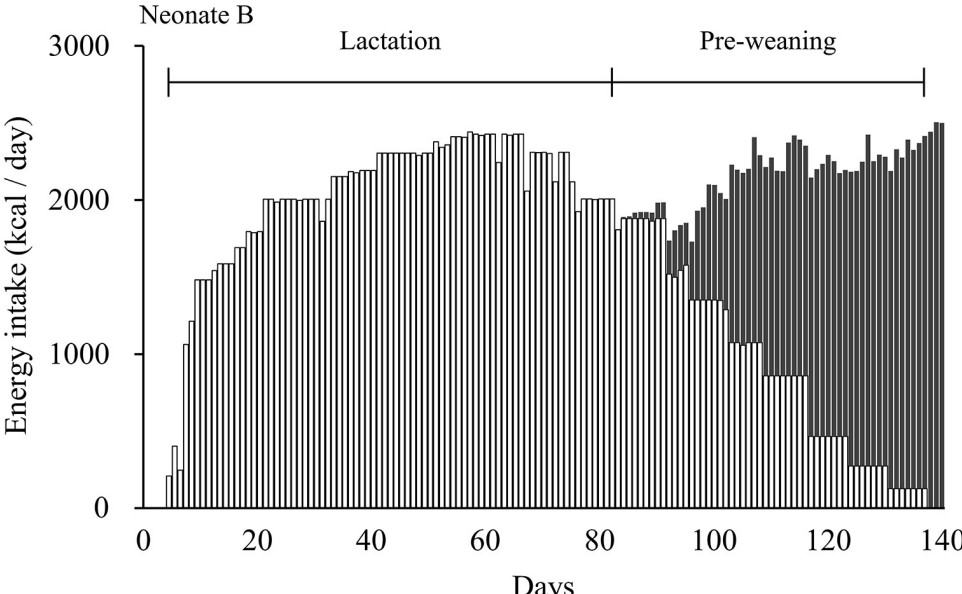

**Fig 3. Daily calorie intake of two neonatal porpoises during the hand-rearing period.** The upper panel shows data for animal A, and the lower panel shows data for animal B. White bars indicate the daily calorie intake of artificial formula, whereas black bars indicate the daily calorie intake of whole fishes. Lactation, neonates fed only artificial formula; Pre-weaning, neonates fed artificial formula and whole fish. (*) Hyperlipidemia; arrowhead indicates time frequent regurgitation was observed in animal A.

during the lactation period (Fig 3). The daily total calorie intake for animal B during the pre-weaning period was significantly ($P<0.01$) lower than that during the lactation period.

In this study, the serum concentrations of total protein, lipid and lipoprotein components, and 40 free amino acids were measured as indicators of nutritional status in the two hand-

**Table 2. Results of hematological and serum biochemical analyses of two hand-reared neonate finless porpoises.**

| Neonate A | | Day | Week | | Month | | | | HTG |
|---|---|---|---|---|---|---|---|---|---|
| **Day** | | 5 | 2 | 3 | 1 | 2 | 3 | 4 | Day 14–18 |
| WBC | ( / μl) | 9200 | 7400 | 8900 | 8900 | 13200 | 17500 | 7100 | |
| RBC | (10⁵ / μl) | 576 | 587 | 578 | 564 | 469 | 491 | 504 | |
| Hb | (g / dl) | 18.5 | 19.3 | 18.9 | 18.2 | 14.9 | 14.5 | 14.4 | |
| Ht | (%) | 56.9 | 57.3 | 56.5 | 55.4 | 44.0 | 44.6 | 45.7 | |
| TP | (g / dl) | 6.3 | 5.7 | 4.5 | 5.3 | 5.6 | 5.8 | 6.1 | |
| Tcho | mg / dl | 157 | 271 | 208 | 195 | 173 | 152 | 213 | 250 |
| TG | mg / dl | 15 | 870 | 1027 | 574 | 362 | 300 | 256 | 1050 |
| PL | mg / dl | 189 | 401 | 344 | ND | 264 | 243 | 283 | 372 |
| FFA | mEq / l | 1325 | 7764 | 3510 | ND | 2043 | 4660 | 1355 | 6890 |
| Glucose | mg / dl | 127 | 128 | 102 | 119 | 100 | 66 | 132 | |
| BUN | mg / dl | 37.1 | 52.8 | 56.6 | 64.0 | 62.5 | 54.5 | 81.7 | |
| **Neonate B** | | Day | Week | | Month | | | | HTG |
| **Day** | | 4 | 2 | 3 | 1 | 2 | 3 | 4 | Day 11–18 |
| WBC | ( / μl) | 6000 | 8400 | 5300 | 13400 | 13200 | 16000 | 17200 | |
| RBC | (10⁵ / μl) | 549 | 561 | 561 | 515 | 458 | 482 | 492 | |
| Hb | (g / dl) | 20 | 20 | 19.7 | 18.0 | 14.2 | 15.0 | 14.5 | |
| Ht | (%) | 60.4 | 60.6 | 58.4 | 52.5 | 42.9 | 44.6 | 43.8 | |
| TP | (g / dl) | 6.0 | 5.4 | 5.2 | 5.5 | 6.5 | 6.4 | 6.7 | |
| Tcho | mg / dl | 144 | 289 | 231 | 157 | 146 | 174 | 186 | 250 |
| TG | mg / dl | 60 | 732 | 477 | 161 | 184 | 569 | 159 | 604 |
| PL | mg / dl | 164 | 362 | 290 | 215 | 233 | 259 | 237 | 330 |
| FFA | mEq / l | 985 | 2958 | 7301 | 1701 | 2083 | 4063 | 1143 | 5129 |
| Glucose | mg / dl | 61 | 89 | 126 | 101 | 72 | 81 | 95 | |
| BUN | mg / dl | 40.5 | 28.2 | 42.7 | 51.5 | 49.4 | 59.6 | 61.0 | |

WBC, white blood cells; RBC, red blood cells; Hb, hemoglobin; Ht, hematocrit; TP, serum total protein; Tcho, total cholesterol; TG, triglyceride; PL, phospholipid; FFA, free fatty acid; BUN, blood urea nitrogen; HTG, hypertriglyceridemia.

reared neonatal porpoises. Results of hematological and serum biochemistry analyses for the two hand-reared porpoises are shown in Table 2. Hypoproteinemia and hypertriglyceridemia were observed in the two neonate porpoises during the early stage of hand-rearing. The serum total protein concentration in animal A decreased from 6.3 g/dl on Day 5 to 4.5 g/dl on Day 18. Hypoproteinemia in animal A improved, and the serum protein concentration increased to 5.8 g/dl by Day 89. Hypertriglyceridemia was observed in animal A between Days 14 and 18. The number of total peripheral leukocytes in animal A increased markedly (14,833 ± 2,329/μl) between Days 75 and 89, when the animal had esophagitis.

Hypoproteinemia was also observed in animal B between Days 11 and 18 and improved by Day 77. Hypertriglyceridemia was also observed in animal B between Days 11 and 18. The number of total peripheral leukocytes in animal B increased markedly after Day 32 (14,044 ± 2,213/μl).

The concentrations of 24 free amino acids in the plasma of the neonate porpoises were significantly ($P<0.05$) higher than those in adult porpoises, and the concentrations of 2 free amino acids were significantly ($P<0.05$) lower (Fig 4). The plasma 3-MH level in the neonates was 195.7 ± 74.0 nmol/ml, which was significantly ($P<0.05$) lower than that of adult porpoises. Plasma valine, leucine, and isoleucine levels in the neonates were 683.4 ± 184.8, 344.9 ± 103.4,

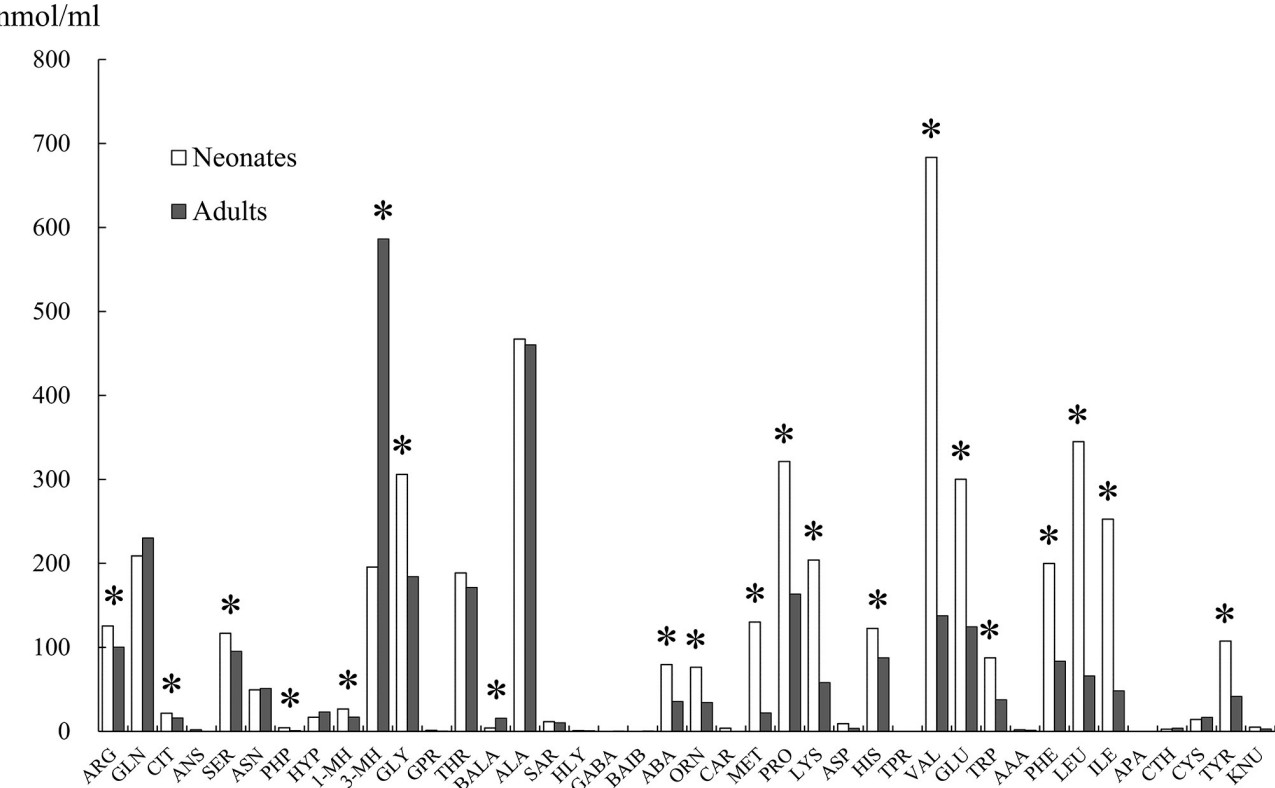

**Fig 4. Plasma free amino acid concentrations in hand-reared neonatal and adult porpoises.** ARG, arginine; GLN, glutamine; CIT, citrulline; ANS, anserine; SER, serine; ASN, asparagine; PHP, prolyl-hydroxyproline; HYP, hydroxyproline; 1-MH, 1-methylhistidine; 3-MH, 3-methylhistidine; GLY, glycine; GPR, glycyl-proline; THR, threonine; BALA, beta-alanine; ALA, alanine; SAR, sarcosine; HLY, hydroxylysine; GABA, gamma-amino butyric acid; BAIB, beta-aminoisobutyric acid; ABA, alpha-aminobutyric acid; ORN, ornithine; CAR, carnosine; MET, methionine; PRO, proline; LYS, lysine; ASP, aspartic acid; HIS, histidine; TPR, thioproline; VAL, valine; GLU, glutamic acid; TRP, tryptophan; AAA, alpha-aminoadipic acid; PHE, phenylalanine; LEU, leucine; ILE, isoleucine; APA, aminopimelic acid; CTH, cystathionine; CYS, cysteine; TYR, tyrosine; KNU, kynurenine. *Significant difference ($P<0.05$).

and 252.6 ± 79.9 nmol/ml, respectively, which were significantly ($P<0.05$) higher than those in adult porpoises. The plasma branched-chain amino acid (BCAA) level in the neonates was 1280.8 ± 360.9 nmol/ml, which was three times higher than that of adult porpoises.

Hyperlipoproteinemia, characterized by a higher percentage of very-low-density lipoprotein (VLDL) and the appearance of midband, was also observed in the two neonates when they had hypertriglyceridemia (Fig 5). A significant correlation ($P<0.01$) was found between serum TG and serum FFA concentrations in the two neonates.

## Discussion

This is the first report of successful complete hand-rearing of endangered finless porpoises. The highest mortality rate in *ex situ* neonatal finless porpoises typically occurs within the first 15 days following birth. The data obtained in this study are indispensable for determining the conditions under which artificial hand-rearing enables newborn porpoises to grow throughout the hand-rearing period. Lactation in dolphins is characterized by small amounts at each breastfeeding and a large number of daily lactations [4,12–14]. In this study, 9.7-kg and 9.8-kg newborn porpoises grew in the first week through 15 daily nursings comprising a total volume of 773.4 ± 190.4 ml (mean ± SD). In the case of animal A, the number of nursings was

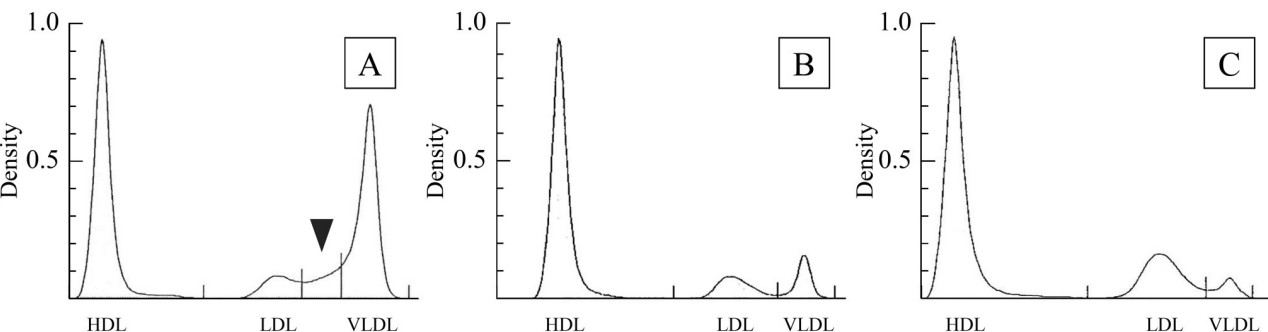

**Fig 5. Densitometric scans of serum lipoproteins in hand-reared neonatal and adult porpoises determined by polyacrylamide gel electrophoresis.**
A, Hand-reared porpoises with hyperlipidemia; B, normal hand-reared porpoises; C, adult porpoises. Arrowhead, midband.

markedly reduced on Day 100 because reflux esophagitis was observed in the area near the cardia. It is possible that frequent tube feeding may damage the mucous membranes of the esophagus. The insertion of a stomach tube into the forestomach of a porpoise and the amount of each tube feeding should be closely monitored when tube feeding is used in artificial hand-rearing.

Analyses of neonatal growth in this study showed that the weight gain of the two calves was not consistent throughout the hand-rearing period. The rate of weight gain declined between 3 and 4 months in the two hand-reared calves. In the case of hand-reared bottlenose dolphins, calf weight gain was shown to increase after 2 months of age and to decrease after 4 to 7 months [4,12]. The day when weight gain declined in the porpoise calves was earlier than that reported for bottlenose dolphins. Porpoise calves begin to eat whole fish spontaneously at approximately 90 days of age (unpublished data). A reduction in the growth rate of captive bottlenose dolphin calves would suggest weaning serves as a response to the increase in energy requirement that the calf could not fulfil through nursing [12,15]. The day when the weight gain of calves declined was similar to the day when naturally nursed porpoise calves begin to spontaneously eat whole fish. The tendency for weight gain at approximately 13 weeks of age suggests that the weaning process begins at that age in porpoise calves.

In this study, the relationship between weight gain and daily calorie intake were examined in hand-reared porpoise calves. Daily calorie requirements in hand-reared bottlenose dolphins and harbor porpoises are reportedly 150 and 200 kcal/kg/day, respectively [2,3]. The mean daily calorie requirement in animals A and B in the present study during the lactation period was 155.9 ± 35.6 and 159.9 ± 20.1 kcal/kg/day, respectively, similar to that of hand-reared bottlenose dolphins. The daily calorie intake in the two calves did not differ between the lactation period and pre-weaning period. Also, the growth rate of the calves in the pre-weaning period was lower than that in the lactation period.

Weight gain reportedly begins to decline at approximately 4 months of age in bottlenose dolphin calves that have started eating whole fish [4]. It is thought that porpoise calves need to grow sufficiently before they can spontaneously catch and eat fish. In addition, it is assumed that the growth rate of calves starts to slow at approximately 13 weeks of age when the weaning process begins.

The highest rate of mortality in bottlenose dolphin calves reportedly occurs within the first 30 days following birth [16]. There are only a few reported cases of successful hand-rearing and only limited clinicopathological data available regarding neonatal cetaceans [4,16]. The improvement in survivability can be attributed to better neonatal and maternal monitoring

and intervention husbandry techniques [4,16]. Few reports have described complete hand-rearing of cetacean neonates beginning immediately from birth [4,17–19]. Data regarding serum biochemistry parameters assessing nutritional status in neonatal dolphins are needed. Hematology and serum chemistry analyses are essential for assessing health status of not only mother porpoises but also neonates in order to improve the survivability of calves [5].

Hypoproteinemia and hypertriglyceridemia were observed in the neonatal porpoises at between 2 and 3 weeks. These characteristics have not been reported in the limited number of cases of hand-rearing of bottlenose dolphins [4,16]. Decreasing serum protein levels are generally observed with malnutrition, along with protein-losing nephropathies and gastrointestinal diseases such as parasitism and protein-losing enteropathies [20]. Hypoproteinemia in marine mammals has been reported in a few limited cases involving rescued harbor seal (*Phoca vitulina*) pups, consistent with malnourishment, and in the case of a rehabilitated bottlenose dolphin [21,22]. In our previous research, we found severe hypoproteinemia in a Commerson's dolphin (*Cephalorhynchus commersonii*) with protein-losing enteropathy [23]. However, it is unlikely that the hypoproteinemia in the neonatal porpoises in the present study was related to small intestinal or renal protein-losing disease. It has been reported that the protein content of dolphin milk is higher than that of other terrestrial mammals, but the reason for this has not been explained [24]. Hypoproteinemia in newborn porpoises suggests that the source of protein needed for neonatal growth has been lacking.

Adequate dietary intake of amino acids is necessary for the growth, development, and health of animals both *in situ* and *ex situ*; thus, more information regarding diets and nutrient profiles of these animals is needed [25]. There are few reports concerning amino acids and their functions in dolphins [6,25,26]. The plasma concentrations of 24 free amino acids in the neonates of the present study were significantly higher than those in adult porpoises. Previous studies have demonstrated that skeletal muscle proteolysis increases in response to energy deficits resulting from malnutrition. Plasma 3-MH levels increase with skeletal muscle proteolysis; thus, 3-MH is generally used as a marker of muscular proteolysis [27,28]. Although there was a decrease in weight gain at approximately 13 weeks of age, the lower plasma 3-MH levels in the neonates compared with adult porpoises indicated that no energy deficit had occurred in the porpoise calves. The temporary hypoproteinemia observed early in the hand-rearing of neonatal porpoises that was not accompanied by significant clinical symptoms is interesting. Serum total protein consists mainly of albumin synthesized in the liver. In rapidly growing newborns, liver function might be overloaded, which in turn can affect protein synthesis. Neonates require a specific balance of amino acids based on growth stage, and amino acid requirements differ between neonatal and adult pigs, for instance [29,30]. Protein synthesis and muscle growth can be stimulated in neonatal pigs by providing amino acids with anabolic properties [29]. These results suggest that neonatal porpoises require protein and supplementation with certain appropriate amino acids in the early stages of hand-rearing. Supporting early muscle growth with an adequate supply of amino acids may be essential for proper growth of neonatal porpoises. We believe that this is an important finding for future case studies and could lead to the development of an ideal artificial milk.

It is worth noting that the plasma BCAA levels of the neonates were 3 times higher than those of adults. Muscle and plasma concentrations of BCAAs are generally elevated by increased physical activity. There is a significant increase in the rate of amino acid catabolism during exercise [31]. BCAAs have been implicated as being involved in energy supply processes in muscle after their release from the liver [32]. It has also been reported that BCAAs are preferentially metabolized by skeletal muscle and that BCAA metabolism in skeletal muscle increases during energy-demanding situations [33]. Increasing BCAA levels during energy restriction can support gluconeogenesis, maintain whole-body and muscle protein synthesis,

and attenuate whole-body and muscle proteolysis [33]. Neonatal porpoises must swim to follow their dam immediately after birth. It is possible that skeletal muscle in newborn porpoises effectively utilizes BCAAs during the early stage of the lactation period. The present study suggests that BCAAs are required for the growth of newborn porpoises.

Hypertriglyceridemia and hypoproteinemia were observed in the neonatal porpoises at between 2 and 3 weeks. It is generally accepted that the fat content of dolphin milk collected from the dam is 26.5%, higher than that of many other mammals [2,34,35]. It is necessary to increase the fat content of artificial formula by adding salmon oil or safflower oil [2,3]. However, the oils are not added to artificial formula to the same concentration as in mother's milk [2]. The fat content of artificial formula is generally 9.0–9.5% [2,3].

Monitoring of serum lipid components and lipoproteins is important for proper nutritional husbandry of captive dolphins [10]. The basis for the addition of oil to an artificial formula and for evaluating neonatal serum lipids have not been established for hand-reared dolphins. Hypertriglyceridemia in bottlenose dolphins has been linked to liver diseases [36]. In this study, midband lipoprotein was observed in serum from the neonates, which developed hypertriglyceridemia in the early stage of hand-rearing. Midband lipoproteins arise from the intermediate products of VLDL catabolism and rarely appear in normal blood [10,37,38]. Hypertriglyceridemia reportedly occurs in human neonatal diabetes and in equine neonates [39,40]. The appearance of midband in blood samples from the hand-reared neonate porpoises with hypertriglyceridemia in the present study suggests the presence of a lipid metabolism disorder. In humans, midband lipoproteins in patients with hyperlipidemia are regarded as indicators of abnormal lipoprotein metabolism in relation to atherosclerosis and cerebrovascular diseases [37,38]. Prolonged hypertriglyceridemia during hand-rearing can cause fatty liver changes and lipid metabolism disorders in neonatal porpoises. These data suggest it is necessary to assess serum protein, free amino acids, lipids, and lipoprotein levels and to feed artificial milk formula to newborn porpoises that is appropriate for their growth stage.

## Conclusion

A method for hand-rearing managed finless porpoises was successfully established in this study. The daily number and amount of nursings and details regarding formula preparation were determined in this case of complete hand-rearing of porpoises. Given the observed hyperlipidemia in the early stages of porpoise hand-rearing, it is possible that there is not a high demand for lipid supplementation in the form of fish oil. The results of this study suggest that neonatal porpoises have a high demand for amino acids, and we found that we needed to add BCAAs to artificial formula for the neonatal porpoises. Nutritional evaluations of serum protein, free amino acids, and lipid components are needed to further improve the survivability of hand-reared neonatal finless porpoises.

The hand-rearing method established in the present study is an essential technique for the husbandry of finless porpoises and can be applied to the conservation of other members of the porpoise family, including vaquita and Yangtze finless porpoises, which are the most endangered dolphins in the world.

## Supporting information

**S1 File.**
(XLSX)

## Acknowledgments

We would like to thank the staff of Toba Aquarium who participated in the hand-rearing of two neonatal finless porpoises. Additionally, we would like to express our gratitude to Atsushi Seko, Yoshihiro Ishihara, and Yoshihito Wakai for professional advice regarding our research.

## Author Contributions

**Conceptualization:** Masahiko Kasamatsu, Masami Furuta.

**Data curation:** Masahiko Kasamatsu, Hiroki Inoue.

**Formal analysis:** Masahiko Kasamatsu, Hiroki Inoue.

**Funding acquisition:** Hidetomo Iwano.

**Investigation:** Masahiko Kasamatsu.

**Methodology:** Masahiko Kasamatsu, Ikuo Wakabayashi, Hiroki Inoue.

**Project administration:** Masahiko Kasamatsu, Masami Furuta.

**Resources:** Masami Furuta.

**Supervision:** Hidetomo Iwano.

**Validation:** Masahiko Kasamatsu.

**Visualization:** Masahiko Kasamatsu.

**Writing – original draft:** Masahiko Kasamatsu, Hidetomo Iwano.

**Writing – review & editing:** Masahiko Kasamatsu, Kazuhiro Hasegawa, Hiroki Inoue, Hidetomo Iwano.

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
