## [Decision Letter · Decision Letter 0]

13 Dec 2023

PONE-D-23-20946The first complete hand-rearing of two neonatal finless porpoisesPLOS ONE

Dear Dr. Iwano,

Thank you for submitting your manuscript to PLOS ONE. After careful consideration, we feel that it has merit but does not fully meet PLOS ONE’s publication criteria as it currently stands. Therefore, we invite you to submit a revised version of the manuscript that addresses the points raised during the review process.

First of all, thank you very much for your patience in this lengthy review process. With respect to the comments by Reviewer 1 concerning statistics, I realize that it may not be possible to address all of them by repeating analyses. If you can do so, that would be excellent, but if you are unable to generate new or additional data due to lack of materials or other problems, please explain your inability to follow through with the requested change.  Please respond to all comments made by the reviewers,

We look forward to receiving your revised manuscript.

Kind regards,

Ulrike Gertrud Munderloh, Ph.D.

Academic Editor

PLOS ONE

Journal Requirements:

Reviewers' comments:

Reviewer's Responses to Questions

**Comments to the Author**

1. Is the manuscript technically sound, and do the data support the conclusions?

Reviewer #1: Yes

Reviewer #2: Yes

2. Has the statistical analysis been performed appropriately and rigorously? 

Reviewer #1: No

Reviewer #2: Yes

3. Have the authors made all data underlying the findings in their manuscript fully available?

Reviewer #1: Yes

Reviewer #2: Yes

4. Is the manuscript presented in an intelligible fashion and written in standard English?

Reviewer #1: Yes

Reviewer #2: Yes

5. Review Comments to the Author

Reviewer #1: This paper reported two cases of successful hand-rearing finless porpoise, which provides invaluable technique, information and data on rescuing the orphans of small cetacean species. This paper mainly provided raw data of their work only with t-test for difference analysis. However, it might be more valubale and helpful for other researchers. So I sugges to further improve the statistical analysis for a better quaility of this work. This paper provided the detailed and supportive data and information on hand-rearing the two finless porpoise babies, which include the formula, the way of hand-rearing, the daily intake, and body weight and blood information of the animals, etc. The process of hand-rearing and nutrition status of the animals were all clearly presented in this paper. But the language needs to be further improved by native speakers.

Reviewer #2: This manuscript provides incredibly valuable data about the hand rearing process for porpoise calves, which as the authors say can be extrapolated to endangered species conservation. I commend the authors for sharing as much data and information as they have. Several comments to consider:

-Lines 23 and 108: consider clarifying "neglect" with "maternal neglect"

-Throughout: consider replacing "force-feeding" with "gavage" or "tube feeding".

-Throughout: consider replacing "lactations" with "nursings" or "feedings", as lactations implies the production by the dam as opposed to feeding by calf.

-L 44-45: consider adding locations for aquaria (not sure where Point Aquarium is).

-L 52: two types of formula are listed, but not discussed or detailed further (the differences between the two).

-L 83: This sentence doesn't provide a whole lot of value since all cetaceans are listed. Consider rephrasing as "N asiaorientalis is listed as endangered..."

-L 92: Given that a main purpose of the study is to evaluate the nutritional status, consider strengthening conclusions about what the nutritional status of these two animals is based on the data collected.

-L 108: consider replacing "breastfeeding" with "nursing".

-L 112: I understand that 'neonate A' changes to 'calf A' as the animal ages into that age class, but it may be unclear to readers that this is the same animal; consider changing to 'animal A/B'.

-L 130: The fact that the slurry of fish meat was fed separately from formula is an important detail, and Table 1 may benefit from separating the slurry line out for clarity.

-Table 1: Consider clarifying "Initial" and "Lactation" as those words don't quite capture the differences between the two (perhaps "Day 1-10" and "Day 11-X", etc.)

-L 155: Consider adding total number of days of complete weaning process (I assume 136-88).

-L 199: confirming vomiting (versus regurgitation)

-L 253-255 and 258-260: listing the mean concentrations of all of these parameters is dense and confusing; consider adding column to Table 2 listing mean values there.

-L 301: unsure why 'grow in the first week' is there; this information appears valuable throughout the entire hand-rearing process; recommend clarification.

-L 310 and 312: "13 and 17 weeks" and "2 months and 4-7 months" - consider standardizing units (weeks or months) for clarity.

-L 352: would argue that this is not just necessary for zoo animals, but all animals (including in situ conservation).

-L 355: the authors have not adequately described why higher plasma amino acid concentrations are associated with hypoproteinemia. Also, is there information to cite that this is an unexpected finding for neonates versus adults (e.g. not formula-raised; is this pathologic, or expected)?

-L 362: Apologies, as above, it is not clear why reduced protein synthesis would occur if amino acid levels were elevated.

-L 382: "bases" - I believe this might be "basis".

-L 388: as above, are there citations that suggest that this is a pathologic metabolism disorder based on what is known in other neonate species? E.g. that hypertriglyceridemia is not expected in nursing calves of the same age?

-L 391: citation needed for "fatty liver changes and lipid metabolism disorders in neonatal porpoises"

-L 399: same as with L 355 above: please explain why hypoproteinemia was associated with elevated amino acid levels in the porpoises.

6. PLOS authors have the option to publish the peer review history of their article (what does this mean?). If published, this will include your full peer review and any attached files.

Reviewer #1: No

Reviewer #2: **Yes: **Claire A Simeone

---

## [Author Response · Author response to Decision Letter 0]

10 Mar 2024

Reviewer #1

This paper reported two cases of successful hand-rearing finless porpoise, which provides invaluable technique, information and data on rescuing the orphans of small cetacean species. This paper mainly provided raw data of their work only with t-test for difference analysis. However, it might be more valubale and helpful for other researchers. So I sugges to further improve the statistical analysis for a better quaility of this work. This paper provided the detailed and supportive data and information on hand-rearing the two finless porpoise babies, which include the formula, the way of hand-rearing, the daily intake, and body weight and blood information of the animals, etc. The process of hand-rearing and nutrition status of the animals were all clearly presented in this paper. But the language needs to be further improved by native speakers.

 In response to your kind comments and constructive suggestions, we would like to express our sincere appreciation. Your thorough examination of our research and your valuable suggestions are highly appreciated. 

 I understand and acknowledge the significance of your suggestion regarding the statistical analysis of our research. But it is difficult to obtain additional hand-rearing data sets from neonatal porpoises because the breeding and hand-rearing of porpoises are extremely rare. We would like to increase the number of cases in the future and verify the data as statistically meaningful. Even though the number of cases in this study is small, we consider it important that this report will be valuable in the conservation of endangered animals, and we would like to deliver it to researchers around the world as soon as possible. 

 Our manuscript has been edited in English by native speakers again. 

Thank you again to the reviewers for their comments and constructive advice. We hope that the changes that have been made to the manuscript meet to your satisfaction.

 We wish to take this opportunity to thank you for your consideration of our paper for publication in your journal, PLOS ONE.

Reviewer #2

 In response to your kind comments and constructive suggestions, we would like to express our sincere appreciation. Your thorough examination of our research and your valuable suggestions are highly appreciated. 

Lines 23 and 108: consider clarifying "neglect" with "maternal neglect". 

 →"neglect" have been changed to "maternal neglect" as suggested (revised manuscript (Rev. M.), line 23 and line 111). 

Throughout: consider replacing "force-feeding" with "gavage" or "tube feeding". 

 →"force-feeding" has been changed to "tube feeding" as suggested (Rev. M., throughout the manuscript). 

Throughout: consider replacing "lactations" with "nursings" or "feedings", as lactations implies the production by the dam as opposed to feeding by calf. 

 →"lactations" have been changed to "nursings" as suggested (Rev. M., throughout the manuscript). 

L 44–45: consider adding locations for aquaria (not sure where Point Aquarium is). 

 →Locations for aquaria have been added as suggested (Rev. M., line 44–45). 

L 52: two types of formula are listed, but not discussed or detailed further (the differences between the two). 

 →The additional information regarding two types of formula has been added as follow: “Safflower oil in the April formula and salmon oil in the Sea World formula have been used for the hand-rearing of dolphins, but whipping cream has only been used in the Sea World formula [1,2]. (Rev. M., line 54). 

L 83: This sentence doesn't provide a whole lot of value since all cetaceans are listed. Consider rephrasing as "N asiaorientalis is listed as endangered...". 

 →This sentence has been rephrased as “is listed as endangered species along with all other cetaceans” as suggested (Rev. M., line 87). 

L 92: Given that a main purpose of the study is to evaluate the nutritional status, consider strengthening conclusions about what the nutritional status of these two animals is based on the data collected. 

 →The sentence strengthening conclusions has been added as suggested (Rev. M., line 411). As follow: “Given the observed hyperlipidemia in the early stages of porpoise hand-rearing, it is possible that there is not a high demand for lipid supplementation in the form of fish oil. The results of this study suggest that neonatal porpoises have a high demand for amino acids, and we found that we needed to add BCAAs to artificial formula for the neonatal porpoises.” 

L 108: consider replacing "breastfeeding" with "nursing". 

 →"breastfeeding" has been changed to "nursing" as suggested (Rev. M., line 112 and 119). 

L 112: I understand that 'neonate A' changes to 'calf A' as the animal ages into that age class, but it may be unclear to readers that this is the same animal; consider changing to 'animal A/B'. 

 →“neonate A”and “neonateB” have been changed to “animal A” and “animal B”, respectively as suggested (Rev. M., throughout the manuscript). 

L 130: The fact that the slurry of fish meat was fed separately from formula is an important detail, and Table 1 may benefit from separating the slurry line out for clarity. 

 →Table 1 has been rearranged as suggested (Rev. M., line 145). 

Table 1: Consider clarifying "Initial" and "Lactation" as those words don't quite capture the differences between the two (perhaps "Day 1-10" and "Day 11-X", etc.). 

 →The specific number of days have been added in Table 1 as suggested (Rev. M., line 145). 

L 155: Consider adding total number of days of complete weaning process (I assume 136-88). 

 →Total number of days of the complete weaning process were have been added as suggested (Rev. M., line 158). 

L 199: confirming vomiting (versus regurgitation). 

 →“vomiting” has been changed to “regurgitation” as suggested (Rev. M., line 202, line 234 and line 247). 

L 253-255 and 258-260: listing the mean concentrations of all of these parameters is dense and confusing; consider adding column to Table 2 listing mean values there. 

 →The mean concentrations of all of these parameters have been deleted and the mean values have been listed in Table 2 (Rev. M., line 261). 

L 301: unsure why 'grow in the first week' is there; this information appears valuable throughout the entire hand-rearing process; recommend clarification. 

 →We agree with the comments from Reviewer #2. The sentences have been added as follows: “The highest mortality rate in ex situ neonatal finless porpoises typically occurs within the first 15 days following birth. The data obtained in this study are indispensable for determining the conditions under which artificial hand-rearing enables newborn porpoises to grow throughout the hand-rearing period.” (Rev. M., line 300). 

L 310 and 312: "13 and 17 weeks" and "2 months and 4-7 months" - consider standardizing units (weeks or months) for clarity. 

 →"13 and 17 weeks" have been changed to "3 and 4 months" as suggested (Rev. M., line 312). 

L 352: would argue that this is not just necessary for zoo animals, but all animals (including in situ conservation). 

 →“of zoo animals” has been changed to “of animals both in situ and ex situ” as suggested (Rev. M., line 355). 

L 355, L 362: the authors have not adequately described why higher plasma amino acid concentrations are associated with hypoproteinemia. Also, is there information to cite that this is an unexpected finding for neonates versus adults (e.g. not formula-raised; is this pathologic, or expected)? Apologies, as above, it is not clear why reduced protein synthesis would occur if amino acid levels were elevated.

 →We would like to discuss hypoproteinemia and plasma amino acid profiles in neonatal porpoises separately. 

The sentences have been added as follows: “The temporary hypoproteinemia observed early in the hand-rearing of neonatal porpoises that was not accompanied by significant clinical symptoms is interesting. Serum total protein consists mainly of albumin synthesized in the liver. In rapidly growing newborns, liver function might be overloaded, which in turn can affect protein synthesis. Neonates require a specific balance of amino acids based on growth stage, and amino acid requirements differ between neonatal and adult pigs, for instance [29,30]. Protein synthesis and muscle growth can be stimulated in neonatal pigs by providing amino acids with anabolic properties [29]. These results suggest that neonatal porpoises require protein and supplementation with certain appropriate amino acids in the early stages of hand-rearing. Supporting early muscle growth with an adequate supply of amino acids may be essential for proper growth of neonatal porpoises. We believe that this is an important finding for future case studies and could lead to the development of an ideal artificial milk. (Rev. M., line 364). 

L 382: "bases" - I believe this might be "basis". 

 →"bases" has been revised to “basis” following your comments (Rev. M., line 393). 

L 388: as above, are there citations that suggest that this is a pathologic metabolism disorder based on what is known in other neonate species? E.g. that hypertriglyceridemia is not expected in nursing calves of the same age? 

 →Yes, the hypertriglyceridemia has not been found in sera from nursing calves of dolphins. Hypertriglyceridemia has been reported to occur in human neonatal diabetes and in equine neonates (Rev. M., line 398). 

L 391: citation needed for "fatty liver changes and lipid metabolism disorders in neonatal porpoises". 

 →The "lipid metabolism disorders" and related citation as suggested have been additionally described from L392 to L403. 

L 399: same as with L 355 above: please explain why hypoproteinemia was associated with elevated amino acid levels in the porpoises. 

 →“Conclusion” section has been revised as suggested and the sentence has been added as follows: “The results of this study suggest that neonatal porpoises have a high demand for amino acids, and we found that we needed to add BCAAs to artificial formula for the neonatal porpoises.” (Rev. M., line 413). 

Thank you again to the reviewers for their comments and constructive advice. We hope that the changes that have been made to the manuscript meet to your satisfaction.

 We wish to take this opportunity to thank you for your consideration of our paper for publication in your journal, PLOS ONE.

---

## [Editor Report · Decision Letter 1]

28 Mar 2024

The first complete hand-rearing of two neonatal finless porpoises

PONE-D-23-20946R1

Dear Dr. Iwano,

We’re pleased to inform you that your manuscript has been judged scientifically suitable for publication and will be formally accepted for publication once it meets all outstanding technical requirements.

Kind regards,

Ulrike Gertrud Munderloh, Ph.D.

Academic Editor

PLOS ONE